# Dissecting the Molecular Role of *ADIPOQ* SNPs in Saudi Women Diagnosed with Gestational Diabetes Mellitus

**DOI:** 10.3390/biomedicines11051289

**Published:** 2023-04-27

**Authors:** Amal F. Alshammary, Sabah Ansar, Raed Farzan, Sarah F. Alsobaie, Arwa A. Alageel, Malak Mohammed Al-Hakeem, Imran Ali Khan

**Affiliations:** 1Department of Clinical Laboratory Sciences, College of Applied Medical Sciences, King Saud University, Riyadh 11433, Saudi Arabia; aalshammary@ksu.edu.sa (A.F.A.); sansar@ksu.edu.sa (S.A.); rfarzan@ksu.edu.sa (R.F.); salsobaie@ksu.edu.sa (S.F.A.); aaalageel@ksu.edu.sa (A.A.A.); 2Department of Obstetrics and Gynecology, College of Medicine, King Khalid University Hospital, Riyadh 11451, Saudi Arabia; malhakeem@ksu.edu.sa

**Keywords:** *ADIPOQ* gene, rs1501299, rs17846866, rs2241766 and GDM women

## Abstract

The traditional definition of gestational diabetes mellitus (GDM) is the leading cause of carbohydrate intolerance in hyperglycemia of varying severity, with onset or initial detection during pregnancy. Previous studies have reported a relationship among obesity, adiponectin (*ADIPOQ*), and diabetes in Saudi Arabia. *ADIPOQ* is an adipokine that is produced and secreted by adipose tissue involved in the regulation of carbohydrate and fatty acid metabolism. This study investigated the molecular association between rs1501299, rs17846866, and rs2241766 single-nucleotide polymorphisms (SNPs) in *ADIPOQ* and GDM in Saudi Arabia. Patients with GDM and control patients were selected, and serum and molecular analyses were performed. Statistical analyses were performed on clinical data, Hardy Weinberg Equilibrium, genotype and allele frequencies, multiple logistic regression, ANOVA, haplotype, linkage disequilibrium, as well as MDR and GMDR analyses. The clinical data showed significant differences in various parameters between the GDM and non-GDM groups (*p* < 0.05). In GDM women with alleles, genotypes, and different genetic models, the rs1501299 and rs2241766 SNPs showed a strong association (*p* < 0.05). Multiple logistic regression analysis revealed a negative correlation (*p* > 0.05). This study concluded that rs1501299 and rs2241766 SNPs were strongly associated with GDM in women in Saudi Arabia.

## 1. Introduction

One of the conditions in gestational diabetes mellitus (GDM) is defined as a glucose/carbohydrate intolerance initially detected during pregnancy and which reconciles after delivery [1]. The overall prevalence of GDM is estimated to be 1.7–11.6% [2]. This transient form of diabetes is estimated to affect maternal pathophysiology [3]. In 1964, O’Sullivan and Mahan established the oral glucose tolerance test (OGTT) for diagnosing GDM [4], which has both short- and long-term health consequences and a global public health burden [5]. GDM is a common risk factor for pregnancy, affecting 30% of the global ethnic population within the past two decades [6]. Being overweight and advanced maternal age, as well as having a family history of dysglycemia, are established risk factors for the development of GDM. Moreover, women with GDM are known have a tenfold increase in risk of developing type 2 diabetes mellitus (T2DM) [7,8]. Furthermore, GDM causes maternal, fetal, and neonatal complications, all of which lead to diabetes. Both T2DM and GDM results in impaired insulin secretion and increased insulin resistance as part of their pathogenesis [9]. Complex and multifactorial elements are involved in GDM pathogenesis, which becomes apparent during the third trimester of pregnancy. Insulin sensitivity levels are diminished by up to 70%, and the effects of placental hormones are enhanced. The progression of GDM causes emerging dysfunction in the pancreatic β-cell leading to deficient maternal insulin levels unable to confront elevated demands [10]. Additionally, GDM is associated with modifiable risk factors such as elevated body mass index (BMI), fat-to-healthy diet, physical activity habits, good quality of life, and environmental factors, as well as non-modifiable risk factors such as family history, inheritance, maternal age, ethnicity, consanguineous marriages/relationships, T2DM in first-degree relatives, and self-history of polycystic ovary syndrome [11]. When the guidelines of the International Association of Diabetes in Pregnancy study group were used instead of the 1999 World Health Organization recommendations (WHO), the prevalence of GDM was found to be 2.4 times higher. Notably, the incidence of GDM appears to vary seasonally, with higher rates observed in summer and lower rates in winter [12]. In most cases, GDM can be controlled through diet or exercise.

The prevalence of GDM In Saudi Arabia is between 8–19% [13]. According to WHO estimates, the prevalence of diabetes is expected to increase by 183% between 2003–2023 [14]. According to Lee et al., Saudi Arabia is the third most populated nation with a prevalence of 22.9% of GDM [15], while Badakhsh et al. estimated the prevalence of GDM in Saudi Arabia at 17.6% [16]. Obesity and T2DM are common in Saudi Arabia, with 41% of Saudi women being classified as obese [17], while 24.5% of pregnant Saudi women in Jeddah were found to be obese [18]. Obesity is currently considered a negative public health consequence leading to an increased risk of developing cardiovascular metabolic diseases, certain types of cancer(s), and other diseases [19]. The combination of maternal obesity and GDM contributes to pregnancy-related metabolic complications connected to fetal overgrowth and adiposity [20]. Adipokines and adipocytokines play significant roles in glucose homeostasis during pregnancy [21,22], and obesity is associated with an increased production of adipokines and fatty acids, which increases the risk of developing GDM. In fact, GDM is consistently associated with adipokines, insulin resistance, and deterioration of glucose tolerance [23]. Adipokines and adipocytokines are secreted by the adipose tissue, and the derived hormone adiponectin (*ADIPOQ*) has insulin-sensitizing, anti-inflammatory, and anti-atherogenic properties, which enhance glucose absorption in muscles and decrease glucose production in the liver [24].

Single-nucleotide polymorphisms (SNPs) are the most common and prevalent types of genetic variation present in humans and refer to single-nucleotide changes at specific sites in the genome. Ample evidence suggests that SNPs can be used to identify individuals at risk for GDM [25]. In this study, we selected the *ADIPOQ* gene because it is linked to GDM via lower adiponectin levels during pregnancy and increased insulin resistance and glucose intolerance later in life, thereby increasing the risk of diabetes [26]. Adiponectin is a 244 amino acid protein found in humans, located on chromosome 3q27. The location on the chromosome has three exons and spans approximately 17 kb. It has been identified as a susceptibility locus for obesity, metabolic syndrome (MetS), T2DM, GDM, and other human diseases [27,28,29,30,31]. This 30 kDa protein is found in adipose tissue that acts as a complement receptor, which was initially identified in 1995 and named adiponectin [32]. The SNPs in the *ADIPOQ* gene have been studied in a limited number of diseases in the Saudi population [33,34,35,36,37], but no studies have been conducted on Saudi women diagnosed with GDM. Based on previous studies and its role in GDM, we chose the rs1501299, rs17846866, and rs2241766 SNPs in the *ADIPOQ* gene for this study, with the aim of investigating the association between these three SNPs in the *ADIPOQ* gene in women with GDM in Saudi Arabia.

## 2. Materials and Methods

### 2.1. Selection of GDM Women

This hospital-based, case-controlled study was designed at King Khalid University Hospital, King Saud University, Riyadh, Saudi Arabia. We selected 110 patients confirmed to have GDM and 110 patients who did not have the condition (control/non-GDM) all treated at the Department of Obstetrics and Gynecology on the university premises. The initial enrollment in this study was based on signing an informed consent form prior to participation. The second criterion was the enrollment of all Saudi women, as this study was designed as a Saudi-based rather than a population-based study. All pregnant women underwent a glucose challenge test (GCT), followed by an OGTT. All 220 Saudi women were enrolled in the GCT and OGTT tests, which were performed based on our previous studies [38]. The complete details for performing both GCT and OGTT tests were shown in detail, and >50% of the OGTT showed elevated levels of glucose, confirming the presence of GDM in pregnant women, and all normal glucose levels obtained were confirmed to belong to the non-GDM (control) patients. Pregnant women who developed diabetes before pregnancy were excluded. The study was conducted in accordance with the principles of the Declaration of Helsinki.

### 2.2. Data Collection from Pregnant Women

Data regarding the anthropometric, biochemical, clinical, and demographic details of the patients with GDM and the control patients were collected. After the patients provided written consent, 5 mL of peripheral blood was extracted from all 220 study participants and separated into two samples, 2 mL for DNA isolation and 3 mL for serum biochemical analyses. Trained nurses from the outpatient clinic collected blood samples and completed the questionnaire from their records. Age, weight, height, BMI, and hypertension status were recorded, as were systolic blood pressure (SBP) or diastolic blood pressure (DBP). Different glucose levels, such as fasting blood glucose (FBG); postprandial blood glucose (PPBG); GCT; OGTT for fasting, 1st hour, 2nd hour, and 3rd hour, as well as lipid profile parameters, such as total cholesterol (TC), triglycerides (TG), high-density lipoprotein cholesterol (HDLc), and low-density lipoprotein cholesterol (LDLc), were measured in the serum samples. Glycated hemoglobin (HbA1c) levels were measured in ethylenediaminetetraacetic acid (EDTA) blood, which was also used to extract genomic DNA. A calorimetric method with an automated chemistry analyzer was used to analyze the serum samples, including the lipid profile parameters.

### 2.3. EDTA Blood Analysis

Molecular analysis was initiated by extracting genomic DNA from 2 mL of EDTA blood. Blood, serum, and genomic DNA were stored at −80 °C. The DNA was quantified using a NanoDrop (Thermo Fisher). Subsequently, DNA amplification was performed for rs1501299, rs17846866, and rs2241766 SNPs in *ADIPOQ* using polymerase chain reaction (PCR). The process starts with 50ng DNA, 10pmoles of the forward and reverse primers, and DNA master mix. Purified H_2_0 was added to complete a 50 µL reaction volume. The PCR program consisted of denaturation (95 °C for 5 min and 30 s), annealing (62 °C–64 °C–66 °C), and extension (72 °C for 45 s and 5 min). The PCR reaction was carried out between 1.25 min 1.30 min for three SNPs and was subsequently visualized on a 2% agarose gel. To digest the PCR products, *BsmI*, *HgaI*, and *SmaI* restriction enzymes were used for 18 h at 37 °C. Specific band sizes were formed based on the design of primers and restriction enzymes (Table 1). The digested PCR products were run on a 3% agarose gel, and images were captured (Figure 1).

### 2.4. Sanger Sequencing Analysis

In this study, rs17846866, rs1501299 rs2241766 SNPs was validated using Sanger sequencing analysis, as we concluded that rs2241766 SNP is very rare in Saudi Arabia as well as in the global population (Figure 2). Sanger sequencing was performed outside the G-141 laboratory at the KSU premises.

### 2.5. Statistical Analysis

In this study, different statistical software was used to perform various tests on patients suffering from GDM and the control patients. The obtained clinical data was converted into categorical (mean ± SD) and numerical variables (total numbers and %). Clinical data of both groups were analyzed using SPSS software (version 27.0, USA), with *p*-values (Table 2). Next, Hardy–Weinberg Equilibrium (HWE) analysis was performed using the control patients for the 3 SNPs with GPower, version 3.1 (Table 3). Both genotype (Table 4) and allele frequencies (Table 5) were calculated using odds ratios (OR), with 95% confidence intervals (95%CI), and *p*-value using the SNPSTAT software. Multiple logistic regression analysis was performed in patients afflicted with GDM between the dependent and independent variables using SPSS software (Table 6). One-way ANOVA (Table 7) was performed using Jamovi software (Version 2.3.21) with Kruskal–Wallis tests. Haplotype analysis was performed (Table 8) using OR, 95%, CI and *p*-values using SNPSTAT software. Furthermore, linkage disequilibrium (LD) for D’ values paired between 3 SNPs using Haploview analysis (Version 4.2) was performed in patients suffering from GDM and the control patients (Figure 3; Table 9). Gene–gene interaction analysis (Table 10) was performed using the generalized multifactor dimensionality reduction (GMDR) model software. Dendrograms (Figure 4) and MDR analyses were performed (Figure 5) for graphical depletion. A bar graph analysis was performed using GraphPad software (version 4.1.2) for family histories of T2DM and GDM among pregnant women confirmed as cases in this study (Figure 6). Statistical significance was confirmed when the *p*-value was less than 0.05 (*p* < 0.05) between two or more groups.

## 3. Results

### 3.1. Details from Questionnaire for Patients with GDM and Control Patients

The demographic and clinical features of the 220 patients (Table 2) was recorded and data obtained from patients not afflicted with GDM were used as a reference for comparison with the data obtained from patients with GDM in the Saudi population. Both significant and statistically significant (*p* < 0.0001) differences were observed regarding age with patients suffering from GDM having an average age of 33.02 years while the control patients had an average age of 29.46 years. The total age of patients with GDM and control patients ranged from 21–45 years and 20–38 years, respectively. There were no significant differences in sex (*p* = 1.00), height (*p* = 0.98), or LDLc (*p* = 0.27). There was a significant difference in anthropometric measurements (weight = 0.0002 and BMI = 0.0001), hypertension levels (SBP and DBP <0.0001), serum glucose levels (FBG, PPBG, GCT, OGTT (F), 1–3 h & HbA1c = <0.0001), lipid profile values (TG = 0.0003, TC = 0.005, and HDLc = 0.001), and family history (T2DM & GDM = < 0.0001). In this study, 7.3% of the women with GDM were on insulin, and the rest were on the recommended diet and completing the physical activity program.

### 3.2. Hardy–Weinberg Equilibrium

The rs1501299, rs17846866, and rs2241766 SNPs were selected for this study. Hardy–Weinberg Equilibrium analysis was performed for the control patients (Table 3). The results showed that rs1501299 was not deviated from HWE (*p* = 0.20). However, the other two SNPs, rs17846866 (*p* = 0.002) and rs2241766 (*p* = 0.001), deviated from HWE.

### 3.3. Calculation of Genetic Frequencies

The genotype frequencies of *ADIPOQ* gene-related SNPs (rs1501299, rs17846866, and rs2241766) in patients with GDM and control patients were calculated (Table 4). The combined genotype frequency of these three SNPs was 100%. The frequency of the rs1501299 SNP in patients with GDM with a heterozygous genotype was found to be 63.6% and in control patients, it was 37.3%, whereas the frequencies of GG and AA were found to be 20% and 16.4% in patients with GDM and 11.8% and 50.9% in control patients, respectively. A strong association was found in AC vs. AA: OR-5.31 (2.76–10.24); *p* < 0.0001, CC vs. AA: OR-5.26 (2.21–12.53); *p* < 0.0001, AC + CC vs. AA: OR-5.30 (2.83–9.94); *p* < 0.0001, and AA + AC vs. CC: OR-2.95 (1.70–5.09); *p* < 0.0001. The genotype frequencies of the rs17846866 SNP were found to be 75.5%, 17.3%, and 7.3% as in the TT, TG, and GG genotypes in women with GDM and 80%, 14.5%, and 5.5% in control patients. None of the association was found either with genotypes (TG vs. TT: OR-1.26 (0.61–2.61); *p* = 0.71 and GG vs. TT: OR-1.41 (0.47–4.25); *p* = 0.73) or any of the genetic models (TG + GG vs. TT: OR-1.30 (0.69–2.46); *p* = 0.42, TT + GG vs. TG: OR-1.36 (0.46–4.06); *p* = 0.58 and TT + TG vs. GG: OR-1.23 (0.59–2.53); *p* = 0.58). For the rs2241766 SNP, the genotype frequencies for TT, TG, and GG genotypes were 70%, 19.1%, and 10.9%, respectively, in women with GDM and 86.4%, 10%, and 3.6%, respectively, in control patients. Genotype (TG vs. TT: OR-2.36 (1.07–5.18); *p* = 0.009 and GG vs. TT: OR-3.70 (1.15–11.94); *p* = 0.02), genetic models such as TG + GG vs. TT: OR-2.71 (1.37–5.36); *p* = 0.003 and TT + GG vs. TG: OR-3.24 (1.01–10.40); *p* = 0.03), showed strong association in patients with GDM when compared with control patients.

### 3.4. Calculation of Allelic Frequencies

The allele frequencies of rs1501299, rs17846866, and rs2241766 SNPs were calculated (Table 5). The call rate for allele frequencies was 100%. The A and C alleles were found in 48.2% and 51.8% of patients with GDM and 69.5% and 30.5% of control patients, respectively. Positive association was confirmed for rs1501299 SNP in C vs. A: OR-2.45 (1.66–3.63); *p* < 0.0001. The allele frequencies for the rs17846866 SNP were 84.1% and 15.9% in GDM patients for the T and G alleles respectively, whereas 87.3% and 12.7% were confirmed in non-GDM patients. Negative association was documented for G vs. T: OR-1.29 (0.75–2.23); *p* = 0.41. The prevalence of the T and G alleles in patients with GDM was 79.5% and 20.5%, respectively, whereas in control patients, it was 91.4% and 8.6%, respectively. A strong positive association was shown between G vs. T: OR-2.71 (1.54–4.90); *p* < 0.0001.

### 3.5. Logistic Regression Analysis Studied in GDM Women

Multiple logistic regression analysis was performed (Table 6) for the rs1501299, rs17846866, and rs2241766 SNPs studied as GDM covariates. In this study, 17 covariates were examined, none of which were found to be associated with any one of them.

### 3.6. One-Way ANOVA

The ANOVA results were obtained using the 3 SNPs in GDM patients (Table 7). Post prandial blood glucose was associated with all 3 SNPs (*p* = 0.01). In the rs1501299 and rs17846866 SNPs, OGTT-3rd hour and TC were also commonly associated (*p* = 0.02 and *p* = 0.01, respectively). Furthermore, TG was commonly associated with rs17846866 and rs2241766 SNPs in patients with GDM (*p* = 0.01). For the rs1501299 SNP, HDLc (*p* = 0.01) OGTT 2nd hour (*p* = 0.01) and 3rd hour levels (*p* = 0.02) were associated. However, for the rs17846866 SNP, HbA1c (*p* = 0.01) and OGTT-fasting levels (*p* = 0.01) were associated and with rs2241766, GCT levels (*p* = 0.01). Unfortunately, FBG levels, BMI, and weight were not associated with any SNP (*p* > 0.05). ANOVA confirmed that PPBG had a common association with all three 3 SNPs studied. Among rs1501299 SNP, elevated levels were present among age (33.50 ± 6.89), weight (81.08 ± 10.71), BMI (32.73 ± 3.99), OGGT-F (7.27 ± 2.85), and TC (6.27 ± 1.44). DBP (74.58 ± 3.14), FBG (6.0 ± 1.25), and PPBG (10.17 ± 19.54) were present in AG genotypes and in GG genotypes, with SBP (120.60 ± 11.47); GCT (9.58 ± 1.05); OGGT levels for 1st hour (10.70 ± 1.23), 2nd hour (9.21 ± 1.99), and 3rd hour (10.58 ± 10.080); HbA1c (5.48 ± 0.45); TG (2.43 ± 1.80); HDLc (1.08 ± 0.4); and LDLc levels (4.03 ± 0.8). In rs17846866 SNP, GG genotypes had elevated levels in weight (82.07 ± 14.59); DBP (75.50 ± 3.35); OGGTT-F (7.53 ± 2.23), 1st hour (10.68 ± 0.95) and 2nd hour (9.50 ± 1.05); HDLc (1.01 ± 0.54); and LDLc (3.93 ± 0.91) levels. The GT genotypes had elevated levels for (6.17 ± 1.08), PPBG (16.76 ± 36.61), GCT (9.39 ± 0.98), OGTT-3rd hour (9.20 ± 0.45), HbA1c (5.46 ± 0.70), and TC (6.0 ± 0.36). Age (33.60 ± 5.70), BMI (32.16 ± 4.20), SBP (120.43 ±10.39), and TG (2.44 ± 2.17) levels were found to be high in TT genotypes. In rs2241766 SNP, GG genotypes had five variates of elevated levels such as age (34.08 ± 5.30), weight (80.40 ± 9.50), BMI (32.70 ± 2.70), HbA1c (5.50 ± 0.30), and LDLc (4.10 ± 0.90), and GT genotypes had elevated levels in SBP (120.40 ± 9.91); DBP (74.61 ± 3.84); and OGTT-1st hour (10.90 ± 1.24), 2nd hour (9.30 ± 2.06) and 3rd hour (9.10 ± 7.96). Seven elevated levels were present in FBG (6.0 ± 1.23), PPBG (9.85 ± 18.65), GCT (9.50 ± 7.14), OGTT F (6.49 ± 2.14), TC (5.76 ± 1.34), TG (2.30 ± 2.18), and HDLc (0.96 ± 0.41) in TT genotypes.

### 3.7. Analysis in ADIPOQ Variants

Haplotype analysis was carried out for the rs1501299, rs17846866, and rs2241766 SNPs in women with GDM (Table 8). The combination of the three SNPs in the female/female population of Saudi Arabia allowed for the detection of seven haplotypes. We recorded various haplotype combinations that were tested against seven predictors for the three outcome measures. Haplotypes C-T-T (OR-3.01 [1.73–5.25]; *p* = 0.004), A-T-G (OR-2.26 [1.14–4.48]; *p* = 0.02), C-G-T (OR-4.06 [1.24–13.31]; *p* = 0.02), and C-T-G (OR-16.66 [2.13–130.42]; *p* = 0.008) were significantly associated with GDM.

### 3.8. Linkage Disequilibrium Analysis

Patients with GDM and control patients were both subjected to linkage disequilibrium (LD) analysis for rs1501299, rs17846866, and rs2241766 SNPs, with the latter identifying both GDM and non-GDM by the delta coefficient (D′). However, LD analysis revealed no evidence of these three SNPs (Figure 3). Neither of the two SNPs, present as either L1 or L2, showed any association with either GDM or non-GDM patients (Table 9).

### 3.9. Interaction of Genetic Variants in GDM Women through MDR and GMDR Analysis

In this study, gene–gene interactions were carried out for the rs1501299, rs17846866, and rs2241766 SNPs in women with GDM (Table 10). Here, R1, R2, and R3 represented the rs1501299, rs17846866, and rs2241766 SNPs, respectively, and there was a strong association (*p* < 0.001) with the combinations of R1 (T = 0.6727), R1/R3 (T = 0.6682), and R1/R2/R3 (T = 0.65). Dendrogram analysis (Figure 4) also shows a strong association with the rs1501299, rs17846866, and rs2241766 SNPs in women with GDM, especially between R2 and R3, followed by R1 with R2/R3. Furthermore, the MDR analysis (Figure 5) confirms the graphical representation model in women via r1-r3, as well as the study results, which affirmed a graphical representation of the combined effect of the entire loci models as high- and low-risk groups and statistical interactions determined by MDR.

## 4. Discussion

In this study, we screened rs1501299, rs17846866, and rs2241766 SNPs from the *ADIPOQ* gene among pregnant Saudi women with GDM via serum glucose analyses, such as GCT and OGTT. Subsequently, we discovered six studies on a Saudi population with different SNPs in *ADIPOQ* gene [33,34,35,36,37,39]. Our data revealed no significant association between the rs17846866 SNP and GDM in Saudi women for any of the genotypic, genetic, or allelic associations. However, a strong association was documented between rs1501299 and rs2241766 SNPs in women with GDM with alleles, genotypes, and different genetic models (*p* < 0.05). Multiple logistic regression analysis revealed a negative association (*p* > 0.05). ANOVA showed a common association with PPBG (*p* = 0.01) in three variants and separate association with rs1501299 (OGTT 1–3, TC, and HDLc), rs17846866 (OGTT F & 3rd hour, HbA1c, TG, and TC), and rs2241766 (GCT and TG) variants (*p* < 0.05). The haplotype was associated, but the LD analysis revealed no association (*p* > 0.05). Both MDR and GMDR (dendrogram and depletion) analyses revealed positive relationships (*p* < 0.05). Previous studies have not documented women with GDM in Saudi Arabia. Therefore, the results of the current study are reliable and robust with respect to genotype and demographic features.

Adiponectin comprises 247 amino acids. *ADIPOQ* is one of the diabetic genes found at the 3q27 locus, according to genome-wide linkage scans. The promoter region is made up of 5’UTR, a major sequence motif in Intron-1 [40]. The SNPs rs1501299 [41], rs17846866 [42], and rs2241766 [43] were discovered to be risk factors for T2DM development in specific populations. Genetic studies have established a link between ADIPOQ and insulin resistance as well as its role in the pathogenesis of T2DM. T2DM and GDM are similar because they share common pathophysiology and risk factors [44]. Furthermore, T2DM and obesity are closely related because they are linked to insulin resistance, which contributes to an elevated BMI. Studies have confirmed that obese patients are nearly seven times more likely to develop T2DM and other heart diseases. Under ideal circumstances, the pancreatic β-cells of the islet of Langerhans secrete enough insulin to prevent hypoglycemia despite drops in insulin levels [45]. Previous studies have confirmed the high prevalence of overweight, obesity, and T2DM in various regions of Saudi Arabia [46,47,48,49], indicating a high risk of obesity and T2DM, implying that patients can develop MetS disorders with further complications. ADIPOQ may play a major role, as ADIPOQ has been documented in almost all chronic diseases globally [50,51].

Global SNP studies of *ADIPOQ* in GDM with various variants, such as rs2241766, rs1501299, rs266729, rs12495941, rs182052, rs140531754, and rs17300539, have revealed all forms of association [24,30,52,53,54,55,56,57,58,59,60,61,62,63,64,65,66]. Different SNPs were studied in the *ADIPOQ* gene in women with GDM in various ethnicities, and in a recent study in Thai women with GDM, a couple of SNPs, rs182052 and rs140531754, were studied and confirmed as negative associations [65]. A meta-analysis was carried out between women with GDM and +45T>G (rs2241766), +276G>T, (rs1501299), and −11377C>G SNPs in the *ADIPOQ* gene, and it was confirmed that +45T>G (rs2241766) SNP was associated in nine separate studies globally [67]. Another meta-analysis of GDM using rs266729 confirmed the risk of GDM in Asian and European women and a decrease in the American population [25]. A meta-analysis examined the rs1501299 and rs2241766 SNPs in T2DM and found a positive association, confirming that both variants had an important impact [68]. However, studies have shown conflicting results, which may be attributed to regional and climatic differences. Therefore, these SNPs should not be used as risk factors or biomarkers for GDM.

Previous studies in Saudi Arabia using genetic variants of *ADIPOQ* have examined coronary artery disease (CAD) in T2DM [33], MetS [34], and PCOS [35] and insulin resistance in non-diabetic Saudi women [36], as well as T2DM [37] and colon cancer [39]. In both the Al-Daghri studies, the T45G (rs2241766) and T276G (rs1501299) SNPs were studied, and the T45G SNP was associated with CAD in T2DM patients [33] and MetS [33]. However, the GG genotype in T45G was found to be 2.1% in both studies and was associated only with CAD disease in T2DM patients (*p* = 0.005). In women with PCOS, nine *ADIPOQ* genetic variants were studied; T45G was one of the most commonly studied variants, and the GG genotype was found in 3.7% of women with PCOS and 2.4% of controls (*p* = 0.29). None of the variants were associated with any format [35]. Another study by Mackawy et al. [36] examined the T45G variant in non-diabetic Saudi women diagnosed with insulin resistance. The GG genotype was found in 7.4% of patients, with a significant association (*p* = 0.003). A recent study by Al-Nbaheen [37] was carried out in patients with T2DM with rs17846866 (*p* = 0.004) and rs1501299 (*p* = 0.01) variants. A previous study by Al-Harithy et al. [39] on colon cancer with T45G and G276T variants showed no association (*p* > 0.05). Our study was carried out with three important disease-causing variants: T45G (rs2241766) was strongly associated with 2.3–3.4 times risk in different genotypes and genetic models, and 10.9% of GG genotypes were present in GDM cases and 3.6% in non-GDM women (*p* = 0.02). In previous studies in Saudi Arabia, the prevalence of the GG genotype was found to be 2.1–7.4%, which is very high in our population. To rule out this difference, we validated the rs2241766 variant in 4.1% of the total sample (n = 220). Sanger sequencing analysis was validated in 4.1% of the total samples, that is, the 3 samples of each with TT, TG, and GG genotypes (Figure 2). Disparities in the findings of similar population studies can be attributed to sample size and disease mode.

In this study, 33.7% of women with GDM had family members with GDM, including 48.6% sisters, 40.5% mothers, 5.4% aunts, and 2.7% grandmothers (Figure 6). However, 2.7% of participants had GDM, one of whom was a mother-in-law. In the case of T2DM family history of the women with GDM, it was present in various family members, including their father (48.2%); mother (40.9%); grandmother (4.5%); sister-in-law (3.6%); and brother, sister, and grandfather (0.9%).

One of the major limitations of this study is the omission of screening for the rs17366743 SNP in the *ADIPOQ* gene, which has previously been confirmed as an integrating factor [69]. Missing time of delivery, pregnancy, and neonatal/fetal complications are another limitation of this study. The lack of serum analysis could be one of its major limitations. The strength of this study lies in the screening of the three variants in Saudi women with GDM.

## 5. Conclusions

This study confirmed that both rs1501299 and rs2241766 SNPs were strongly associated with GDM in women in Saudi Arabia. To predict the role of GDM, a large screening program with the majority of SNPs present in the *ADIPOQ* gene is required, particularly for obese women during pregnancy. Serum studies are also strongly advised to investigate insulin sensitivity and resistance in participants.

## Figures and Tables

**Figure 1 biomedicines-11-01289-f001:**
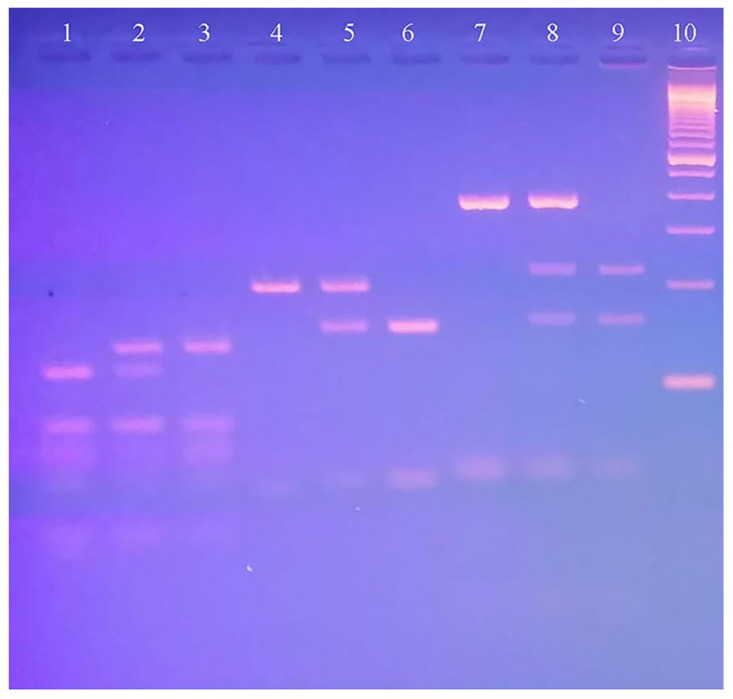
A 3% agarose gel picture shows the digested PCR products with all possible band sizes for rs1501299, rs17846866, and rs2241766 SNPs in the *ADIPOQ* gene in. Lane 1: *ADIPOQ* (rs1501299) Homozygous AA genotype; Lane 2: *ADIPOQ* (rs1501299) Heterozygous AC genotype; Lane 3: *ADIPOQ* (rs1501299) Homozygous CC genotype; Lane 4: *ADIPOQ* (rs17846866) Homozygous TT genotype; Lane 5: *ADIPOQ* (rs17846866) Heterozygous TG genotype; Lane 6: *ADIPOQ* (rs17846866) Homozygous GG genotype; Lane 7: *ADIPOQ* (rs2241766): Homozygous TT genotype; Lane 8: *ADIPOQ* (rs2241766): Heterozygous TG genotype; Lane 9: *ADIPOQ* (rs2241766): Homozygous GG genotype; Lane 10: 100bp DNA ladder.

**Figure 2 biomedicines-11-01289-f002:**
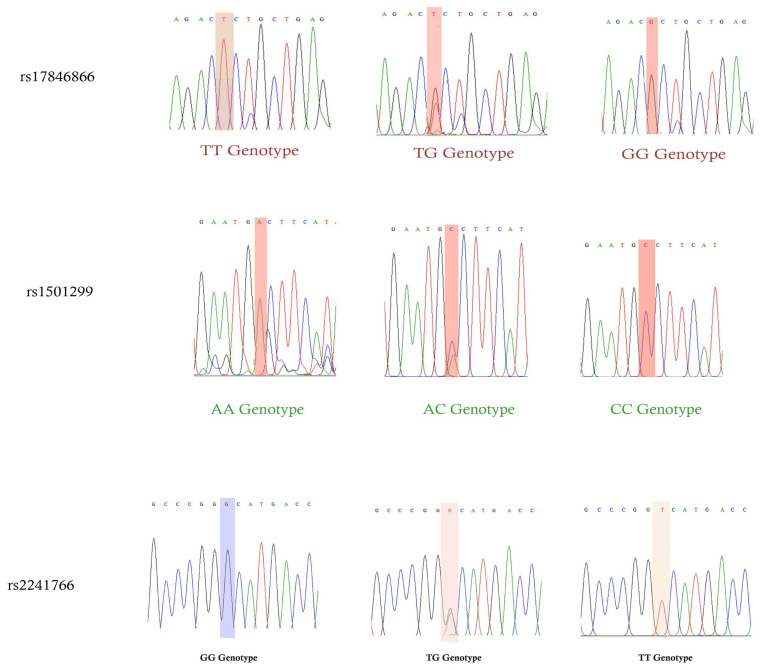
Validation was performed via Sanger sequencing analysis for rs1501299, rs17846866, and rs2241766 SNPs in *ADIPOQ* gene.

**Figure 3 biomedicines-11-01289-f003:**
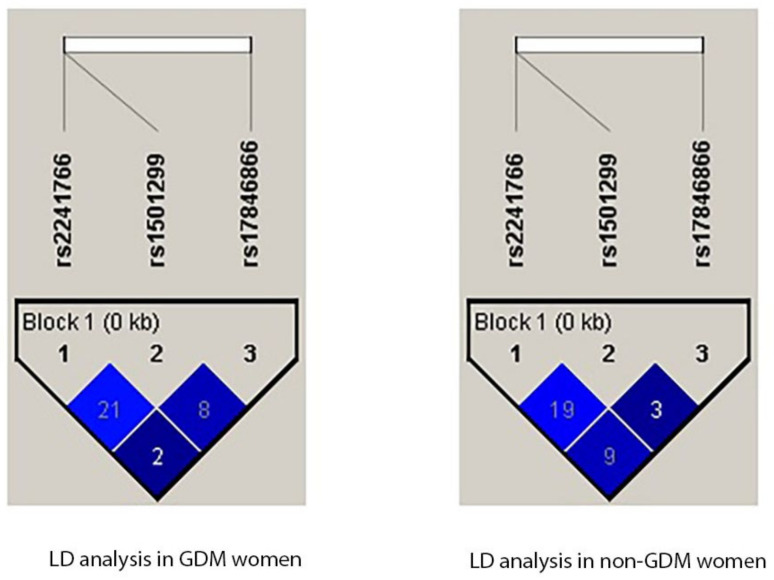
Linkage Disequilibrium analysis was studied for rs1501299, rs17846866, and rs2241766 SNPs in the *ADIPOQ* gene in GDM and non-GDM women.

**Figure 4 biomedicines-11-01289-f004:**
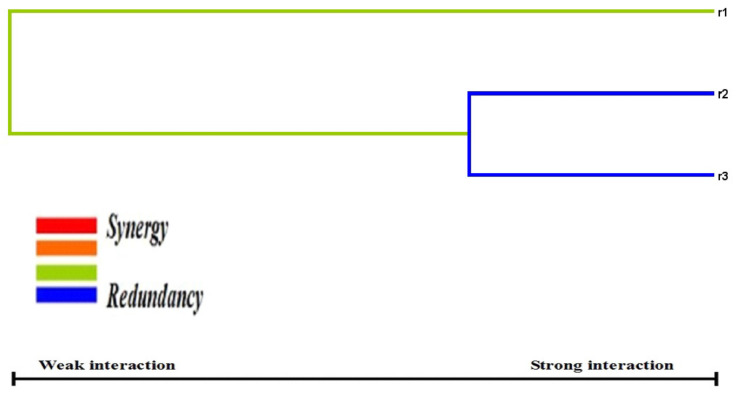
Representation of dendrogram using 3 SNPs involved in GDM patients.

**Figure 5 biomedicines-11-01289-f005:**
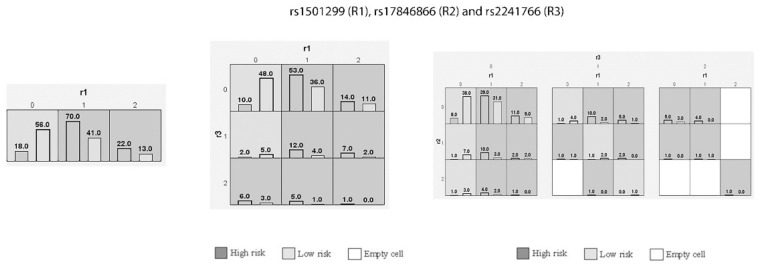
Graphical depiction of Multifactor Dimensionality Reduction analysis regarding GDM. Darker cells indicate riskier combinations, whereas lighter cells indicate the lower-risk group. White/blank cells represent genotype combinations for which no data are available. For each multifactor combination, the bars represent the hypothetical case (left) and control (right) distributions.

**Figure 6 biomedicines-11-01289-f006:**
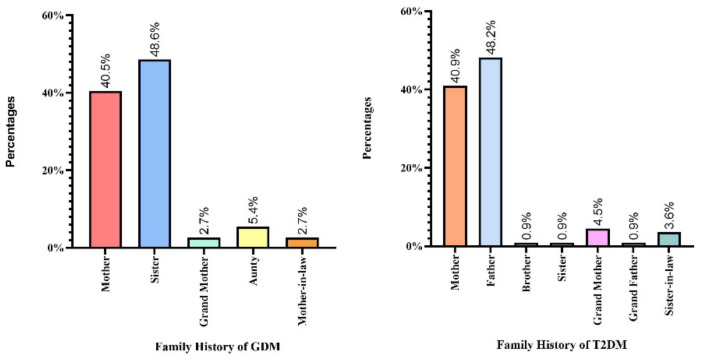
Representation of family histories of GDM and T2DM in GDM patients.

**Table 1 biomedicines-11-01289-t001:** List of *ADIPOQ* SNPs and other details used in the study.

Gene	ADIPOQ	ADIPOQ	ADIPOQ
rs Number	rs1501299	rs17846866	rs2241766
SNPs	A-**C**/(c.276G > C)	T-**G**/(+10211T > G)	T-**G**/(+45T > G)
Location	Intron 2	Intron 1	Exon 2
Forward Primer	GGCCTCTTTCATCACAGACC	GCTAAGTATTACAGATTTCAGGGCAG	GAAGTAGACTCTGCTTGAGATGG
Reverse Primer	AGATGCAGCAAAGCCAAAGT	CAGCAACAGCATCCTGAGC	TATCAGTGTAGGAGGTCTGTGATG
PCR Size	196bp	222bp	372bp
Annealing Temperature	64 °C	66 °C	62 °C
Restriction Enzyme	BsmI (GAATGCC^↑^)	HgaI (GAATGCC^↑^)	SmaI (GGG^↑^CCC)
Digested Products	A-196bp; C-146/50bp	T-222bp; G-115/107bp	T-372bp; G-219/153bp

^↑^ breakpoint for specific restriction enzymes.

**Table 2 biomedicines-11-01289-t002:** Clinical data obtained from all patients involved in the study.

Covariates	Controls (n = 110)	GDM (n = 110)	*p*-Value
Age (year)	29.46 ± 6.16	33.02 ± 5.87	0.0001
Gender (male: female)	0 (0%):110 (100%)	0 (0%):110 (100%)	1.00
Weight (kilograms)	73.14 ± 11.64	79.25 ± 12.32	0.0002
Height (centimeters)	157.92 ± 5.01	157.91 ± 5.49	0.98
BMI (kg/m^2^)	29.29 ± 4.07	31.80 ± 4.42	0.0001
SBP (mmHg)	110.42 ± 11.86	120.15 ± 10.09	<0.0001
DBP (mmHg)	64. 37 ± 3.01	74.40 ± 3.31	<0.0001
FBG (mmol/L)	4.31 ± 0.98	5.93 ± 1.18	<0.0001
PPBG (mmol/L)	4.96 ± 12.32	9.31 ± 15.73	<0.0001
GCT (mmol/L)	6.21 ± 1.58	9.33 ± 1.04	<0.0001
OGTT (F) (mmol/L)	4.92 ± 1.08	6.59 ± 2.14	<0.0001
OGTT (1) (mmol/L)	7.23 ± 1.65	10.61 ± 1.76	<0.0001
OGTT (2) (mmol/L)	6.32 ± 1.47	9.09 ± 1.74	<0.0001
OGTT (3) (mmol/L)	4.13 ± 1.27	6.16 ± 1.79	<0.0001
Hb1Ac (%)	4.78 ± 0.28	5.41 ± 0.34	<0.0001
TC (mmol/L)	5.06 ± 1.14	5.75 ± 1.27	0.0003
TG (mmol/L)	1.71 ± 1.23	2.33 ± 1.95	0.005
Hdlc (mmol/L)	0.71 ± 0.27	0.95 ± 0.41	0.001
Ldlc (mmol/L)	3.71 ± 0.94	3.85 ± 0.97	0.27
Medication (Insulin)	NA	08 (7.3%)	NA
Family History of T2DM	28 (25.5%)	110 (100%)	<0.0001
Family History of GDM	10 (9.1%)	37 (33.7%)	<0.0001

**Table 3 biomedicines-11-01289-t003:** Analysis of Hardy–Weinberg Equilibrium three SNPs in the *ADIPOQ* gene.

SNPs	Minor Allele	Genotype Frequencies for Controls (n = 110)	ꭓ^2^
rs1501299	C	AA-50.9%, AC-37.3% and CC-11.8%	1.58
rs17846866	G	TT-80%, TG-14.6% and GG-5.5%	13.1
rs2241766	G	TT-86.4%, TG-10% and GG-3.6%	14.7

**Table 4 biomedicines-11-01289-t004:** Genotyping analysis for patients with GDM and non-GDM women with three SNPs in the *ADIPOQ* gene.

Gene (rs Number)	Genotypes	GDM (n = 110)	Non-GDM (n = 110)	OR (95%CI) and *p*-Value
*ADIPOQ* (rs1501299)	AA	18 (16.4%)	56 (50.9%)	1.00
	AC	70 (63.6%)	41 (37.3%)	OR-5.31 (2.76–10.24); *p* < 0.0001
	CC	22 (20%)	13 (11.8%)	OR-5.26 (2.21–12.53); *p* < 0.0001
	AC + CC vs. AA	92 (83.6%)	44 (49.1%)	OR-5.30 (2.83–9.94); *p* < 0.0001
	CC + AA vs. AC	40 (36.4%)	69 (71.7%)	OR-1.87 (0.89–3.92); *p* = 0.09
	AA + AC vs. CC	98 (80%)	97 (88.2%)	OR-2.95 (1.70–5.09); *p* < 0.0001
*ADIPOQ* (rs17846866)	TT	83 (75.5%)	88 (80%)	1.00
	TG	19 (17.3%)	16 (14.5%)	OR-1.26 (0.61–2.61); *p* = 0.71
	GG	8 (7.3%)	6 (5.5%)	OR-1.41 (0.47–4.25); *p* = 0.73
	TG + GG vs. TT	27 (24.6%)	22 (20%)	OR-1.30 (0.69–2.46); *p* = 0.42
	TT + GG vs. TG	91 (82.8%)	94 (85.5%)	OR-1.36 (0.46–4.06); *p* = 0.58
	TT + TG vs. GG	102 (92.7%)	104 94.5%)	OR-1.23 (0.59–2.53); *p* = 0.58
*ADIPOQ* (rs2241766)	TT	77 (70%)	95 (86.4%)	1.00
	TG	21 (19.1%)	11 (10%)	OR-2.36 (1.07–5.18); *p* = 0.009
	GG	12 (10.9%)	4 (3.6%)	OR-3.70 (1.15–11.94); *p* = 0.02
	TG + GG vs. TT	33 (30%)	15 (13.6%)	OR-2.71 (1.37–5.36); *p* = 0.003
	TT + GG vs. TG	89 (80.9%)	99 (90%)	OR-3.24 (1.01–10.40); *p* = 0.03
	TT + TG vs. GG	98 (89.1%)	106 (96.4%)	OR-2.12 (0.97–4.65); *p* = 0.054

**Table 5 biomedicines-11-01289-t005:** Allele frequencies between patients with GDM and control patients with three SNPs in the *ADIPOQ* gene in Saudi women.

Gene (rs Number)	Genotypes	GDM (n = 110)	Control (n = 110)	OR (95%CI) and *p*-Value
*ADIPOQ* (rs1501299)	A	106 (48.2%)	153 (69.5%)	Reference
	C	114 (51.8%)	67 (30.5%)	OR-2.45 (1.66–3.63); *p* < 0.0001
*ADIPOQ* (rs17846866)	T	185 (84.1%)	192 (87.3%)	Reference
	G	35 (15.9%)	28 (12.7%)	OR-1.29 (0.75–2.23); *p* = 0.41
*ADIPOQ* (rs2241766)	T	175 (79.5%)	210 (91.4%)	Reference
	G	45 (20.5%)	19 (8.6%)	OR-2.71 (1.54–4.90); *p* < 0.0006

**Table 6 biomedicines-11-01289-t006:** Multiple logistic regression analysis was performed on *ADIPOQ* genetic variants and covariates during an investigation into the association between three SNPs in the *ADIPOQ* gene in GDM women in Saudi Arabia.

Covariates	R-Value ^a^	Adjusted R Square Value	Standardized β-Coefficient for rs1501299	Standardized β-Coefficient for rs17846866	Standardized β-Coefficient for rs2241766	F	*p* Value ^b^
Age	0.130	−0.011	0.018	−0.120	0.052	0.609	0.610
Weight	0.067	−0.024	−0.030	−0.376	−0.051	0.159	0.924
BMI	0.102	−0.018	−0.025	−0.098	−0.003	0.371	0.774
SBP	0.046	−0.026	0.012	−0.045	0.004	0.075	0.974
DBP	0.128	−0.012	−0.068	0.111	−0.007	0.585	0.626
FBG	0.125	−0.012	−0.036	0.052	−0.113	0.559	0.643
PPBG	0.119	−0.014	−0.021	0.104	−0.057	0.506	0.679
GCT	0.192	0.010	0.168	−0.064	−0.055	1.359	0.259
OGTT (F)	0.152	−0.005	0.345	0.341	0.306	0.833	0.479
OGTT (1)	0.125	−0.012	0.285	0.282	0.253	0.565	0.639
OGTT (2)	0.105	−0.017	0.282	0.279	0.250	0.390	0.760
OGTT (3)	0.207	0.016	1.173	1.163	1.043	1.582	0.198
Hb1Ac	0.165	0.000	0.120	−0.054	0.122	0.993	0.399
TC	0.135	−0.010	−0.122	0.065	−0.027	0.657	0.580
TG	0.098	−0.018	0.019	−0.078	−0.055	0.343	0.794
HDLc	0.077	−0.022	0.051	0.047	−0.024	0.213	0.888
LDLc	0.241	0.032	0.171	−0.036	0.195	2.188	0.094

^a^ R value; ^b^
*p* value.

**Table 7 biomedicines-11-01289-t007:** Analysis of variance between 3 SNPs present on the *ADIPOQ gene* with clinical/biochemical parameters assessed in GDM women.

	*ADIPOQ* (rs1501299)				*ADIPOQ* (rs17846866)				*ADIPOQ* (rs2241766)			
	AA (n = 18)	AC (n = 70)	CC (n = 22)	*p*-Value	GG (n = 08)	GT (n = 19)	TT (n = 83)	*p*-value	GG (n = 12)	GT (n = 21)	TT (n = 77)	*p*-Value
Age	33.50 ± 6.89	32.75 ± 5.83	33.5 ± 4.64	0.24	33.25 ± 5.42	30.42 ± 5.77	33.60 ± 5.7	0.98	34.08 ± 5.30	32.85 ± 5.40	32.90 ± 5.98	0.78
Weight	81.08 ± 10.71	78.58 ± 13.15	79.85 ± 10.17	0.28	82.07 ± 14.59	75.27 ± 11.83	79.88 ± 11.91	0.74	80.4 ± 9.50	74.47 ± 14.17	79.98 ± 12.43	0.37
BMI	32.72 ± 3.99	31.45 ± 4.7	32.17 ± 3.51	0.25	31.91 ± 4.71	30.19 ± 4.43	32.16 ± 4.2	0.89	32.70 ± 2.70	30.77 ± 4.68	31.94 ± 4.49	0.15
SBP	120.38 ± 11.66	119.94 ± 9.05	120.6 ± 11.47	0.22	119.37 ± 9.87	119.26 ± 8.35	120.43 ± 10.39	0.53	120.0 ± 10.6	120.4 ± 9.91	120.1 ± 9.98	0.95
DBP	74.44 ± 2.83	74.58 ± 3.14	73.81 ± 3.99	0.25	75.5 ± 3.35	74.73 ± 3.41	74.22 ± 3.24	0.95	74.20 ± 3.20	74.61 ± 3.84	74.37 ± 3.14	0.49
FBG	5.83 ± 1.14	6.0 ± 1.25	5.79 ± 0.91	0.25	5.9 ± 0.88	6.17 ± 1.08	5.88 ± 1.21	0.51	5.50 ± 1.1	5.88 ± 0.93	6.0 ± 1.23	0.31
PPBG	7.87 ± 1.55	10.17 ± 19.54	7.75 ± 1.2	0.01 *	7.17 ± 0.81	16.76 ± 36.61	7.81 ± 1.58	0.01 *	7.4 ± 1.5	8.43 ± 1.62	9.85 ± 18.65	0.01 *
GCT	8.95 ± 0.77	9.37 ± 1.08	9.58 ± 1.05	0.26	9.1 ± 1.12	9.39 ± 0.98	9.35 ± 1.04	0.90	9 ± 0.9	9.34 ± 0.61	9.5 ± 7.14	0.01 *
OGTT (F)	7.27 ± 2.85	6.5 ± 1.96	6.19 ± 1.8	0.06	7.53 ± 2.23	5.61 ± 1.17	6.72 ± 2.21	0.01 *	6 ± 1.9	7.2 ± 2.04	6.49 ± 2.14	0.91
OGTT (1)	10.02 ± 1.47	10.69 ± 1.93	10.70 ± 1.23	0.04 *	10.68 ± 0.95	10.50 ± 1.69	10.61 ± 1.83	0.14	10.3 ± 1.8	10.90 ± 1.24	10.55 ± 1.85	0.11
OGTT (2)	9.11 ± 0.96	9.05 ± 1.79	9.21 ± 1.99	0.01 *	9.50 ± 1.05	8.23 ± 1.69	9.25 ± 1.73	0.30	9.1 ± 1.2	9.3 ± 2.06	9.02 ± 1.69	0.16
OGTT (3)	5.9 ± 1.5	7.96 ± 7.03	10.58 ± 10.08	0.02 *	5.62 ± 1.18	9.20 ± 0.45	8.16 ± 7.49	0.01 *	8.1 ± 7.3	9.10 ± 7.96	7.9 ± 7.16	0.85
Hb1Ac	5.30 ± 0.35	5.40 ± 0.29	5.48 ± 0.45	0.52	5.27 ± 0.3	5.46 ± 0.70	5.41 ± 0.30	0.01 *	5.5 ± 0.3	5.4 ± 0.43	5.39 ± 0.3	0.09
TC	6.27 ± 1.44	5.62 ± 1.23	5.73 ± 1.11	0.01 *	5.76 ± 1.99	6.0 ± 0.36	5.69 ± 1.21	0.01 *	5.7 ± 1.3	5.71 ± 0.87	5.76 ± 1.34	<0.03 *
TG	2.28 ± 0.95	2.32 ± 2.16	2.43 ± 1.80	0.96	2.25 ± 1.15	1.89 ± 0.83	2.44 ± 2.17	0.01 *	2.0 ± 0.6	2.3 ± 1.4	2.3 ± 2.18	0.01 *
Hdlc	1.02 ± 0.41	0.89 ± 0.39	1.08 ± 0.4	0.01 *	1.01 ± 0.54	0.97 ± 0.36	0.94 ± 0.40	0.39	0.9 ± 0.3	0.95 ± 0.41	0.96 ± 0.41	0.45
Ldlc	3.58 ± 1.01	3.86 ± 0.98	4.03 ± 0.8	0.50	3.93 ± 0.91	3.69 ± 0.83	3.88 ± 0.99	0.64	4.1 ± 0.9	4.08 ± 1.09	3.74 ± 0.90	0.53

* Indicates statistical association.

**Table 8 biomedicines-11-01289-t008:** Haplotype association between *ADIPOQ* variants in GDM patients.

S. No	rs1501299	rs17846866	rs2241766	Freq	OR (95% CI)	*p*-Value
1	A	T	T	0.4275	1.00	-
2	C	T	T	0.3087	3.01 (1.73–5.25)	*p* = 0.004
3	A	T	G	0.0852	2.26 (1.14–4.48)	*p* = 0.02
4	A	G	T	0.0718	1.34 (0.56–3.17)	*p* = 0.51
5	C	G	T	0.0466	4.06 (1.24–13.31)	*p* = 0.02
6	C	T	G	0.0354	16.66 (2.13–130.42)	*p* = 0.008
7	C	G	G	0.0207	2.57 (0.66–10.01)	*p* = 0.18

**Table 9 biomedicines-11-01289-t009:** Performance of linkage disequilibrium analysis in GDM and non-GDM patients with three SNPs in the *ADIPOQ* gene.

Subjects	L1	L2	D′	r^2
GDM cases	rs2241766	rs1501299	0.212	0.012
GDM cases	rs2241766	rs17846866	0.028	0.0
GDM cases	rs1501299	rs17846866	0.087	0.001
non-GDM	rs2241766	rs1501299	0.19	0.001
non-GDM	rs2241766	rs17846866	0.094	0.006

**Table 10 biomedicines-11-01289-t010:** Gene–gene interaction to determine the GDM risk with three SNPs in the *ADIPOQ* gene.

Model No	Genes Included in Best Combination in Each Model	Training Accuracy	Testing Accuracy	CVC	*p*-Value
1	rs1501299 (R1)	0.6727	0.6727	10/10	<0.001
2	rs1501299, rs2241766 (R1, R3)	0.6869	0.6682	10/10	<0.001
3	rs1501299, rs17846866, rs2241766 (R1, R2, R3)	0.698	0.65	10/10	<0.001

## Data Availability

Data is not applicable towards this study.

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
