# Peer review of "Dissecting the Molecular Role of ADIPOQ SNPs in Saudi Women Diagnosed with Gestational Diabetes Mellitus"

_biomedicines, 2023, doi:10.3390/biomedicines11051289_

Round 1
Reviewer 1 Report
INTRODUCTION
- Line 42: Gestational Diabetes is a maternal and not a fetal pathology, please rewrite the sentence
- Line 49: Type 2 diabetes mellitus is a diabetic status and not a risk factor for gestational diabetes
- I would like to suggest that we eliminate dissertations on the history and risk factors of gestational diabetes and focus on the object of the study, which is ADIPOQ, i.e., the literature on this.
METHODS
- Line 106: How was the oral glucose loading curve performed ? Why did some women perform the GCT and others the OGTT ? At how many weeks of pregnancy was the test performed ? Are all pregnancies single or also twins ? Please better define the study population.
RESULTS
- Table 2: Pregnancy outcomes are missing (complications, timing of delivery, fetal/neonatal complications, etc.)
Author Response
1st Reviewer comments
INTRODUCTION
- Line 42: Gestational Diabetes is a maternal and not a fetal pathology, please rewrite the sentence
- A) We have updated in the revised manuscript. Thank you for your comment.
- Line 49: Type 2 diabetes mellitus is a diabetic status and not a risk factor for gestational diabetes
- A) Dear Reviewer, we have updated in the revised manuscript.
- I would like to suggest that we eliminate dissertations on the history and risk factors of gestational diabetes and focus on the object of the study, which is ADIPOQ, i.e., the literature on this.
- A) Based on your comment, we have modified the sentence and updated in the revised manuscript.
METHODS
- Line 106: How was the oral glucose loading curve performed ? Why did some women perform the GCT and others the OGTT ? At how many weeks of pregnancy was the test performed ? Are all pregnancies single or also twins ? Please better define the study population.
- A) GCT was performed on all pregnant women who visited the out-patient clinic during the second or third trimester (depending on their feasibility), and pregnant women with positive GCT values were then proceeded to perform OGTT test the following week. All pregnant women should fast for at least 8 hours or overnight fasting with three days of unrestricted diet. Fasting samples were taken from all pregnant women, 100 grams of glucose was given, and blood was drawn every hour for the first, second, and third hours. We had all of the single pregnancies in our study. All of these details were included in the revised manuscript, along with a reference to our preliminary work in GDM [1].
RESULTS
- Table 2: Pregnancy outcomes are missing (complications, timing of delivery, fetal/neonatal complications, etc.)
- A) Unfortunately, all the details were not recorded and we will mention as one of the limitations of this study.
- Alharbi, K.K.; Khan, I.A.; Abotalib, Z.; Al-Hakeem, M.M. Insulin receptor substrate-1 (IRS-1) Gly927Arg: correlation with gestational diabetes mellitus in Saudi women. BioMed research international 2014, 2014.
Reviewer 2 Report
The article "Dissecting the molecular role of ADIPOQ genetic variant studies in Saudi women diagnosed with Gestational Diabetes 3 Mellitus", by Alshammary A at all, is very interesting from a scientific point of view.
The recommendation for increasing the quality of the manuscript are:
1. what is the final conclusion in terms of the clinical approach to your findings? Please develop the clinical importance of the presence of rs1501299 and rs2241766 SNPs and the association with GDM women.
2. please provide the number of the registered protocol of the study.
Author Response
2nd Reviewer comments
Comments and Suggestions for Authors
- Q) The article "Dissecting the molecular role of ADIPOQ genetic variant studies in Saudi women diagnosed with Gestational Diabetes 3 Mellitus", by Alshammary A at all, is very interesting from a scientific point of view.
- A) We really appreciate your comment. We will consider as our hard work for this manuscript was recognized.
- Q) The recommendation for increasing the quality of the manuscript are:
- what is the final conclusion in terms of the clinical approach to your findings? Please develop the clinical importance of the presence of rs1501299 and rs2241766 SNPs and the association with GDM women.
- A) The Adiponectin gene plays an important role in the Saudi population. Previous research has established a strong link between adiponectin and the Saudi population. In our study, we had GDM women with and without obesity, indicating that adiponectin plays a significant role in this study. Previous studies with rs1501299 and rs2241766 SNPs in the Saudi population found a strong role, and we found a significant association in our study as well. However, the most important point to discuss in this study was the presence of homozygous variants in rs2241766 SNP, which was highly visible in this study when compared to previous studies performed in the Saudi population, and to revalidate our data, we repeated the samples with RFLP analysis as well as sanger sequencing.
- please provide the number of the registered protocol of the study.
- A) Biochemical and molecular analysis were carried out in this study. PCR was commonly used for restriction fragment length polymorphism, as well as Sanger sequencing analysis to revalidate results.
Reviewer 3 Report
The work concerns a growing problem, i.e. gestational diabetes, the main determinants of which are the same as those of type 2 diabetes. Therefore, together with environmental conditions, genetic predispositions may contribute to the development of insulin resistance and gestational diabetes. Undoubtedly, a defect in the action of adiponectin may play a role in this pathomechanism, as the relationship between adiponectin and insulin resistance is well known. Adiponectin polymorphisms have already been analyzed in many studies, especially in the context of the development and course of diabetes. Hence the doubt, what new does this work bring in this area? And what clinical implications could this have?
Author Response
3rd Reviewer comments
- Q) The work concerns a growing problem, i.e., gestational diabetes, the main determinants of which are the same as those of type 2 diabetes. Therefore, together with environmental conditions, genetic predispositions may contribute to the development of insulin resistance and gestational diabetes. Undoubtedly, a defect in the action of adiponectin may play a role in this pathomechanisms, as the relationship between adiponectin and insulin resistance is well known. Adiponectin polymorphisms have already been analyzed in many studies, especially in the context of the development and course of diabetes. Hence the doubt, what new does this work bring in this area? And what clinical implications could this have?
- A) Dear Editor, First and foremost, we were impressed by your terrible query. However, our team discussed and reached the following conclusions, and we have all done our best to explain your comment.
To begin with, gestational diabetes mellitus (GDM) and type 2 diabetes mellitus (T2DM) share a pathophysiology. One of the most interesting differences between GDM and T2DM is that GDM is a reversible form of diabetes that develops during pregnancy and resolves after delivery. In the future, 5-10 years after resolving the GDM, the same women are more likely to develop T2DM, whereas in T2DM patients, diabetes develops during adultery (~40 years of age) and is a lifelong disease, implying that it is an irreversible form of diabetes.
Diabetes (especially T2DM) and obesity were found to be prevalent in the Saudi population, with women having a higher prevalence of T2DM and obesity than men. As you mentioned, Adiponectin has been linked to a variety of human diseases, including metabolic syndrome, type 2 diabetes, polycystic ovary syndrome, colon cancer, obese women, and coronary artery disease in type 2 diabetes patients. There have been no studies in Saudi women diagnosed with GDM, and the prevalence of GDM has been increasing in Saudi women for over a decade, based on unrestricted diet, western lifestyle, physical inactivity, genetics, and family histories of specific diseases. The novelty of this study can be attributed to the presence of a high prevalence of homozygous variants in the rs2241766 SNP in GDM women, which was revalidated using Sanger sequencing analysis.
Round 2
Reviewer 1 Report
I can be satisfied with the Authors' responses to the comments therefore in my opinion the manuscript can be evaluated by the Editor for publication.